# The Cholinergic System, the Adrenergic System and the Neuropathology of Alzheimer’s Disease

**DOI:** 10.3390/ijms22031273

**Published:** 2021-01-28

**Authors:** Rola A. Bekdash

**Affiliations:** Department of Biological Sciences, Rutgers University, Newark, NJ 07102, USA; rbekdash@newark.rutgers.edu

**Keywords:** acetylcholine, adrenergic, Alzheimer, cholinergic, cognition, epigenetics, Locus Coeruleus, norepinephrine, signaling

## Abstract

Neurodegenerative diseases are a major public health problem worldwide with a wide spectrum of symptoms and physiological effects. It has been long reported that the dysregulation of the cholinergic system and the adrenergic system are linked to the etiology of Alzheimer’s disease. Cholinergic neurons are widely distributed in brain regions that play a role in cognitive functions and normal cholinergic signaling related to learning and memory is dependent on acetylcholine. The Locus Coeruleus norepinephrine (LC-NE) is the main noradrenergic nucleus that projects and supplies norepinephrine to different brain regions. Norepinephrine has been shown to be neuroprotective against neurodegeneration and plays a role in behavior and cognition. Cholinergic and adrenergic signaling are dysregulated in Alzheimer’s disease. The degeneration of cholinergic neurons in nucleus basalis of Meynert in the basal forebrain and the degeneration of LC-NE neurons were reported in Alzheimer’s disease. The aim of this review is to describe current literature on the role of the cholinergic system and the adrenergic system (LC-NE) in the pathology of Alzheimer’s disease and potential therapeutic implications.

## 1. Introduction

Alzheimer’s disease is a prevalent and progressive neurodegenerative disease that afflicts the aging population worldwide. It is considered a global public health problem with economic and societal ramifications. According to the World Health Organization (WHO), it has been estimated that 115.4 million of people will suffer from this debilitating disease by 2050. The Alzheimer’s Association in the United States estimated that 5.8 million Americans 65 years and older are suffering from Alzheimer’s or other dementias in 2020 and this number could reach 13.8 million by 2050 [1]. Based on data from the National Center for Health Statistics, Alzheimer’s disease is ranked as the sixth-leading cause of death in the United States in 2018 [2]. Despite scientific advances that helped in understanding the pathology of Alzheimer’s disease (AD), an effective long-term treatment to prevent or mitigate its symptoms is still not achieved. Pursuing different strategies in AD treatment should then be the goal moving forward.

Late-stage Alzheimer’s disease (AD) dementia is a multigenic and multifaceted disease characterized by progressive decline in cognitive functions such as memory, learning, thinking abilities, and behavioral abnormalities. The neuropathological features of AD are the accumulation of beta-amyloid plaques and hyperphosphorylated tau, a microtubule-associated protein, in neurofibrillary tangles. These features have been observed in several brain regions such as the nucleus basalis of Meynert (NBM) in the basal forebrain, frontal lobe, hippocampus, cingulate gyrus, amygdala, substantia nigra, several brainstem nuclei and the cerebral cortex [3]. Other neuropathological features that are associated with AD are changes in synaptic signaling, loss of synapses and in some cases neuronal loss or degeneration [4]. It has been reported in several studies that AD brain shows dysregulation of the cholinergic system and the Locus Coeruleus noradrenergic system (LC-NE) in several brain regions [5,6]. Acetylcholine (Ach) is the main neurotransmitter that is synthesized by cholinergic neurons. Cholinergic neurons are widely distributed in the brain and cholinergic signaling has been shown to modulate many cognitive functions such as learning, memory, attention and thinking abilities [7]. Norepinephrine (NE) is the main neurotransmitter that is synthesized by LC-NE neurons, has a neuroprotective role, plays a role in cognitive functions, and has been shown to play a role in neurodegenerative diseases such as AD [8].

The aim of this review is to describe the role of the cholinergic system and the LC-NE system dysregulation in the neuropathology of AD. This manuscript presents current research in humans and animal models on the potential therapeutic role of targeting the cholinergic or the adrenergic system to mitigate the symptoms or slow down the progression of this debilitating disease. We will also address the role of environmental factors such as diet in modulating the expression of key genes related to cognition by epigenetic mechanisms, such as methylation.

## 2. Acetylcholine, Cholinergic System and Alzheimer’s Disease

Cholinergic signaling is dependent on the synthesis and release of the neurotransmitter acetylcholine (Ach). Most of the brain regions that are innervated by cholinergic neurons play a role in learning, memory, stress response, and cognitive functions and the degeneration of these neurons is considered a main factor in the development of dementia including AD [7]. These regions include the basal forebrain and nucleus basalis of Meynert, hippocampus and the cerebral neocortex. Ach has diverse functions on neuronal signaling depending on the site of its release, its affinity to specific types of cholinergic receptors (Ach nicotinic receptors (nAchRs) or Ach muscarinic receptors (mAchRs)), its elimination from synapses by acetylcholine esterase (AchE) and the type of its target neurons. Although Ach has an excitatory effect in the periphery at the neuromuscular junction, it is described as a neuromodulator in the brain as it can cause activation or inhibition in a neuron’s firing depending on the type of environmental stimuli or inputs that the target neuron receives [9].

### 2.1. Acetylcholine and Cholinergic Neurons

Ach is synthesized from choline and Acetyl-Coenzyme A (Acetyl-CoA) in a chemical reaction that is catalyzed by choline acetyltransferase (ChAT). Several studies demonstrated the involvement of Ach in cognitive functions such as attention, thinking abilities, learning and memory [9,10]. Moreover, the levels of ChAT have been shown to be altered with aging and in AD [11]. Once synthesized, Ach is transported by the vesicular acetylcholine transporter (VAchT) and stored in synaptic vesicles. Upon depolarization of the presynaptic neuron, Ach is released into the synaptic cleft where it binds to postsynaptic receptors such as Ach muscarinic receptors (mAchRs) or Ach nicotinic receptors (nAchRs). The excess of Ach in the synaptic cleft is degraded by the activity of acetylcholine esterase (AchE) into choline and acetate. Choline is then recycled and taken back by the presynaptic neurons via choline transporters (mainly CHT1). This reuptake significantly contributes to the pool of choline that will be used for Ach resynthesis by cholinergic neurons. Choline can also be phosphorylated back to phosphotidylcholine, a main cellular membrane component that maintains cellular membrane integrity [12]. Figure 1 illustrates the role of Ach in the brain at synapses.

The “Cholinergic Hypothesis of Alzheimer’s Disease” [13] proposed by Perry at al. in 1999 links the dysregulation of the basal forebrain cholinergic neurotransmission, alteration in the levels of cholinergic markers such as Ach, choline, and ChAT, to the age-related cognitive functions’ decline with AD. This hypothesis is still accepted by the scientific community, however, it has been challenged by some and several other competing hypotheses have then emerged [13,14,15,16]. These emerging hypotheses then suggest that other systems besides the cholinergic system contribute to the pathogenesis of AD. There is no doubt that all these hypotheses and studies so far have increased our understanding of the complexity of this neurodegenerative disease and paved the way for preventive measures for treatment [17]. The cholinergic system plays a pivotal role in the etiology of AD. The degeneration of cholinergic neurons in nucleus basalis of Meynert in the basal forebrain, neurons that project to the neocortex, hippocampus and amygdala, contributes to memory deficits in AD patients [18,19]. The levels of ChAT and AchE, main markers of cholinergic neuronal activity, have been shown to be reduced in late stages of AD [20]. For example, a reduction in the activity of ChAT in the hippocampus, cortex and amygdala of AD patients was reported with lesser reductions in other brain regions. A reduction in AchE activity was also reported in the brain suggesting the contribution of the cholinergic system and the role of Ach deficits in the pathogenesis of AD [21]. Moreover, inhibitors of AchE activity have been shown to improve cognitive functions in AD patients, although the efficacy of these inhibitors as drugs has been shown so far to be limited [20,22].

Where are these cholinergic neurons located in the human brain? The origin of ChAT-expressing neurons was identified as the nucleus basalis of Meynert (NBM) in the basal forebrain [23] with significant loss of these neurons in AD brain suggesting a correlation between the cholinergic system, Ach, NBM and the pathogenesis of AD [18]. Cholinergic neurons of the basal forebrain provide cholinergic innervation to cortical and limbic regions. For example, they send axonal projections to the hippocampus, amygdala and the neocortex [24]. The loss of these basal forebrain cholinergic neurons or their dysfunction has been linked to a decrease in ChAT activity, decrease in Ach release in specific brain regions such as in the hippocampus and cortical areas and to AD-related cognitive dysfunctions [25].

### 2.2. Acetylcholine Receptors and Cholinergic Signaling

Cholinergic signaling is determined by the ability of Ach to bind to respective Ach receptors (AchRs) on target neurons to modulate their excitability. Since the effectiveness of acetylcholinesterase inhibitors as drugs has been shown to be limited, the focus has been shifted toward understanding how the modulation of the activity of AchRs and hence the modulation of Ach levels could lead to promising outcomes in AD treatment [26]. Are these receptors impaired in AD patients? In general, AD is characterized by a reduction in the number of nicotinic and muscarinic receptors in basal forebrain cholinergic neurons [27]. There are two types of AchRs that are widely distributed in the brain, muscarinic (mAchRs) and nicotinic receptors (nAchRs) [25]. There are 5 mAchR types (M1–M5) that induce signaling via different types of G proteins [28]. Twelve different subtypes of nAchR subunits (α2–α7, α9, α10 and β2–β10) have been identified in the brain that could coassemble. Despite this large number of nicotinic acetylcholine receptors, it has been reported that α4β2 nAchRs and α7 nAchRs are the most common nAchRs in the brain [29,30] and important potential targets for drug intervention. Of the two subtypes, α7 nAchR has lower affinity to Ach compared to α4β2 nAchR. In the context of neurodegenerative diseases, reduction in the density of α4β2 nAchRs and α7 nAchRs has been reported in the brain of AD patients and this reduction has been suggested to play a significant role in AD pathogenesis [31,32]. From a physiological perspective, α7 nAchRs activation in microglia plays a role in activating anti-inflammatory pathways and regulators of the oxidative stress [33]. This then suggests the importance of targeting α7 nAchRs for AD treatment where neuroinflammation and oxidative stress are both pathological mechanisms identified in this disease. The other type of receptors, the cholinergic α4β2 nAchRs, are abundantly expressed in the brain and play a role in cognitive functions. Postmortem AD brains show reduction in the expression of these receptors in advanced stages of Alzheimer’s dementia in several brain regions such as the cortex, hippocampus and the basal forebrain and similar reduction in the number of these receptors has been reported in a cohort of patients with mild cases of Alzheimer’s dementia [34].

Could interfering with cholinergic receptors function have beneficial outcomes on the AD brain? Ach and the basal forebrain cholinergic signaling modulate prefrontal cortex (PFC) and sensory cortex circuitry related to attention, and the hippocampal circuitry related to memory, two physiological processes severely impaired in AD patients [25]. The deletion of the β2 subunit of nAchRs in the PFC neurons of the prelimbic region of a mouse model resulted in attention deficit and impairment in attention performance, whereas inducing the expression of these receptors using a lentivirus had positive effects on attention but not on “motivational behavior” [35]. The α7 subunit of nAchRs were first identified to interact with amyloid-beta peptide (Aβ) [36] although the later can also activate the β2 subunit of nAchRs. α7 nAchR is specifically important in AD as it is involved in Aβ internalization and is highly expressed in the striatum, thalamus, neocortex and limbic system indicating a role for this receptor in age-related cognitive decline [37]. Interestingly, the α7 subunit and β2 subunit can coassemble together and form an α7β2 nAchRs. This later is shown to be expressed in the basal forebrain neurons of rodents and humans and in hippocampal interneurons and to be vulnerable to elevated Aβ that leads to toxicity [38,39]. The nAchR-Aβ interaction has been shown to modulate signaling mechanisms related to neuroprotection, synaptic plasticity, learning and memory [40,41,42]. There is no surprise that dysfunction of these receptors is linked to the etiology of AD. A decrease in α7 nAchRs-Aβ complex formation has been shown to improve learning and memory whereas an increase in the formation of this complex negatively impacted cholinergic signaling and cognitive functions [40,43,44]. Hence the use of drugs, agonists or antagonists, that affects the nAchR-Aβ interaction could reduce Aβ-mediated neurotoxicity. Future research is needed to elucidate how this interaction is changing in different individuals at different stages of AD and what is the cellular mechanism involved to come up with better therapeutic approaches for AD treatment. The review by Hoskin et al., 2019, summarizes some of the mechanisms of action of some studied drugs that target the nAchRs in AD and the observed effects [45].

The use of mouse models of AD and studying the role of the cholinergic receptor in cholinergic signaling helped in understanding the pathology of this disease and its outcome on brain functioning. The α7 nAchR is highly expressed in the hippocampus and has been shown to enhance long term potentiation (LTP) suggesting a potential mechanism by which Ach can play a role in memory in the human brain [46]. Cholinergic signaling was also detected in the amygdala, a brain region that plays a role in the consolidation and encoding of emotional memories and α7 nAchR has been shown to induce LTP in the amygdala [47]. Although cholinergic signaling has been linked to the cognitive decline seen in AD patients, the mechanism by which this cholinergic system is impaired is not very well understood. Compounding evidence from human and animal model studies further showed that AchRs are major targets of beta amyloid plaque toxicity. For example, beta amyloid deposition induced glutamate release in the hippocampus via the increase in the activity of α7 nAchR [48]. An increase in nicotine intake in a transgenic mouse model of AD (3xTg-AD) was associated with an upregulation in the expression of nicotinic receptors and aggregation and phosphorylation of tau. This indicates that excess nicotine adversely affects the cholinergic system in this AD mouse model. They also showed a reduction in the expression of α7 nAchR in an age-dependent manner in 3xTg-AD mice compared to control mice in specific brain regions [49]. It has been documented that basal forebrain cholinergic neurons are uniquely reliant on Nerve Growth Factor (NGF) signaling for survival. Implanting encapsulated NGF-producing cells into the basal forebrain of patients with mild or moderate AD resulted in improvement of cognitive functions and in the detection of cholinergic markers in the cerebrospinal fluid such as the cortical nicotinic receptor expression [50]. This shows that maintaining the integrity of cholinergic signaling may mitigate the symptoms associated with AD such as attention and memory-related behaviors.

What about mAchRs? Blockade of mAchR signaling in the hippocampus impairs memory [51]. There are several types of mAchRs that are expressed in the brain. In the context of AD, a reduction in the expression or activity of M1-M4 AchRs in the cortex and the hippocampus of AD patients has been reported [52]. It has been suggested that the accumulation of amyloid plaques in cholinergic neurons may decrease the ability of mAchRs to transmit cholinergic signals and impair cholinergic neurons activity [53]. The use of muscarinic agonists against specific types of mAchRs in a mouse model of AD has shown promising results such as decrease in β-amyloid plaque accumulation, decrease in tau hyperphosphorylation and improvement in cholinergic activity and improvement in specific memory-related tasks [20,54]. M1 and M4 subtypes of mAchRs have gained attention as useful therapeutics for mitigating or treating symptoms of AD. These subtypes are highly expressed in the brain and in key regions related to cognition such as the cortex, hippocampus, and striatum. These receptors are also impacted in AD [55]. Thomsen et al., 2018, stated that the use of genetically engineered knockout mice where we have selective inactivation of one or more of these muscarinic receptors subtypes provided a better understanding of their function in neuronal signaling and behavioral outcomes and the role of Ach in the central nervous system [56]. For example, studies conducted in knockout mice suggested the role of M1 and M4 subtypes of mAchRs in shaping hippocampal circuitry, hippocampal release of Ach and hippocampal functions [57,58]. Another study showed that the lack of M1 AchRs in the mouse PFC impaired cholinergic signaling in pyramidal neurons resulting in behavioral changes such as cue detection deficits in these mice [59]. In transgenic mice with amyloid precursor (APP) or tau mutations, the knockout of M1 mAchRs worsened β-amyloid plaque deposition and resulted in cognitive deficits [60]. M1 agonists in many cases have shown promising results in reducing β-amyloid plaques deposition and in improving cognitive functions [61]. M2 subtype of mAchRs have been shown to play a role in induction of LTP in the hippocampus as M2 knockout mice showed learning deficits and did not do well in memory-related tasks [62]. Table 1 summarizes the findings of select studies that demonstrated the role of cholinergic receptors in the brain.

## 3. Norepinephrine, Adrenergic System and Alzheimer’s Disease

Noradrenergic neurotransmission is dependent on the release of the neurotransmitter norepinephrine (NE) at synapses. NE plays an important role in behavior and cognitive functions such as attention, learning and memory [63,64]. NE has a neuroprotective role against chronic neuroinflammation in a rodent Parkinson’s disease model [65] and possibly a general neuroprotective effect against neurodegeneration [66,67]. There is strong evidence derived from human studies and animal models that links the dysfunction of the adrenergic system to the etiology of AD [68,69]. The Locus Coeruleus (LC) is the main noradrenergic nucleus and the main site for the synthesis of NE in the brain. The LC-NE system consists of neurons that project to different brain regions and supplies NE to the cortex, hippocampus, striatum, amygdala, cerebellum, basal forebrain, and the hypothalamus [70]. LC, NE, and adrenergic receptors (ARs) play a significant role in AD pathology and in Aβ peptides production and secretion. In this section, we will focus on the role of the Locus Coeruleus (LC) and forebrain noradrenergic signaling in cognitive functions and how the dysfunction of the LC-NE system is linked to cognitive and behavioral deficits often seen in AD patients. We will also address the role of ARs in modulation of noradrenergic circuitry in AD.

### 3.1. Norepinephine and LC-NE Neurons

Misfolded hyperphosphorylated tau and degeneration of LC neurons have been detected in the neuropathology of AD and that these changes in fact occur early and may contribute to the progression of the disease [8,71]. Moreover, postmortem AD brains show abnormalities in the LC neurons that range from structural changes, neuronal loss to decreases in the levels of noradrenergic markers [72,73,74,75,76,77,78,79]. LC neurons degeneration is linked to Aβ-induced neurotoxicity suggesting the adverse effects of Aβ deposition on LC neurocircuitry [6].

The catecholamine NE is synthesized from tyrosine via a series of enzymatic reactions. Tyrosine hydroxylase (TH) is considered a marker of noradrenergic neurons as it catalyzes the main reaction that converts tyrosine to L-DOPA, the main precursor for dopamine. NE is then synthesized from dopamine via the catalytic activity of dopamine-β-Hydroxylase (DBH) and is destroyed or eliminated in synapses via the activity of monoamine oxidase (MAO) and catechol-O-methyltransferase (COMT). NE can then be recycled back to the presynaptic neuron by the NE transporter (NET), a major determinant of NE levels in the synaptic cleft. The noradrenergic system regulates many physiological processes such as visceral, cognitive, behavioral, motor control, and attention [80]. Figure 2 illustrates the role of NE in the brain at synapses.

The LC neurons constitute 50% of the noradrenergic system and the main neurons that are affected in AD with a loss of around 70% of these neurons in AD patients [81]. This loss was evidenced by the decrease in the activity and plasma levels of DBH during early stages of AD [82]. This may suggest that the use of NE reuptake inhibitors could be a selective treatment in early stages of AD to mitigate the loss of LC neurons activity. Other studies reported a decrease in the level of dopamine in the cerebrospinal fluid (CSF) of AD patients [83] and a reduction of dopamine in the hippocampus and insular cortex of mouse model of AD [84]. COMT polymorphism may have provided an explanation for the psychosis often seen in AD [85]. Although NE levels have been shown to decrease in advanced cases of AD, its levels are elevated or even not changed in other AD cases [77]. This indicates that AD pathology is much more complicated and not simply due to NE input to different brain regions [86]. It also suggests that the level of NE in the brain is dependent on the stage of AD. Early stage shows an elevation of NE levels in AD brain as a compensatory mechanism whereas with the late stage of the disease, NE levels decline due to progressive loss or degeneration of NE-producing neurons.

### 3.2. Adrenergic Receptors and Adrenergic Signaling

LC consists of a heterogeneous population of NE-producing neurons that project to other brain regions. LC-NE system is considered a major hub that communicates the PFC to other brain regions [87]. NE exerts its modulatory effects by binding to ARs on target cells, which are members of the G-protein coupled receptor (GPCR) family. Three main types of ARs are expressed in the brain such as α1, α2 and β with several subtypes of each and varying affinities to NE in the brain and different physiological functions due to different G-proteins and activation of different downstream proteins throughout the brain. It has been shown that α2 ARs have the highest affinity to NE, followed by α1 ARs then β Ars [88]. All of these receptors play an important role in maintaining normal cognitive functions in humans and in animal models [88]. For example, abnormal receptor densities and changes in secondary messengers or effectors that act downstream of the AR signaling have been demonstrated in AD brains providing clear evidence that the malfunction of ARs plays a role in the etiology of AD, such as cognitive dysfunctions [89,90,91,92]. There is supporting evidence in animal models and human studies that shows that α1 ARs play a pivotal role in synaptic plasticity, memory and cognitive functions and have been associated with AD. Long term depression (LTD) was induced by α1 ARs even when the majority of NE innervation to the hippocampus was reduced via the application of NET or MAO inhibitors on rat hippocampal slices or simply by inducing the degeneration of LC-NE neurons using a neurotoxin [93]. This study then suggests that α1 ARs are activated in AD and that this activation is often manifested by malfunctioning of NE neurotransmission in the hippocampus. Perez, 2020 described the role of α1 ARs in short-term and long-term synaptic plasticity in different brain regions such as the hippocampus, neocortex and the PFC, regions adversely affected in AD patients [70].

Blockade of α1 ARs or the use of an antagonist has shown improvement in cognitive deficits and improvement in memory deficits over time in APP23 transgenic mice models of AD [94]. Interestingly, the dysregulation of the LC-NE system during early stages of AD has been linked to Aβ peptide accumulation at LC neuronal terminals. This accumulation has been shown to be linked to β2 and α2 ARs, that is, the binding of Aβ to these receptors and modulation of synaptic transmission in specific brain regions. For example, β2 ARs have been shown to alter amyloid precursor protein (APP) processing [95] and to mediate Aβ-induced tau pathology in the prefrontal cortex of APP/PS1 transgenic mouse model [96]. In addition, Aβ42 accumulation has been shown to induce β2ARs internalization and degradation in primary prefrontal cortical neurons leading to malfunctioning of adrenergic activities in these neurons [97]. This indicates that β2AR activities and levels are quite reduced in AD and are prime factors in LC-NE dysregulation and in the influence of that dysregulation on cognitive functions. The α2 ARs have been also shown to regulate APP proteolytic processing and Amyloid Aβ (Aβ) production and secretion in the cerebral cortex of AD transgenic mice model [98]. These findings then suggest that adrenergic receptors are also influenced by Aβ deposition, and hence impact the release of NE and modulate LC-NE circuitry in AD. This means that pharmacological intervention at early stages of AD, that could alter the interaction of ARs with Aβ, could decrease or delay LC neurodegeneration in AD.

The role of NE in the hippocampus for normal cognition has been reported. For example, immunotoxic ablation of LC-NE neurons in the hippocampus impaired working memory using the water maze task test and impaired neurogenesis by reducing the proliferation but not the survival or differentiation of neural progenitor cells in the dentate gyrus of young rats [99]. Other studies showed that activation of specific β AR in the hippocampus of experimental animal models has a role in contextual and spatial memory consolidation and retrieval [100,101,102]. The administration of blockers for β AR in amyloid-beta protein precursor (APP) mouse model impaired cognitive behavior including in wild-type mice. Interestingly, chronic administration of this beta-blocker enhanced peripheral and central inflammation. This suggests that chronic application of beta blockers in AD should be carefully considered because of its potential inflammatory reaction in the brain [103]. Collectively, these studies highlight the role of ARs in learning and memory and their potential to act as therapeutic targets for treatment of AD.

What about other markers of LC neurons’ activity? Animal models of AD and postmortem AD brain tissues showed a decreased number of tyrosine hydroxylase-producing LC neurons, neurons that mediate cognitive functions such as attention, memory and arousal [104]. This degeneration and cell death of LC neurons were attributed to the accumulation of neurofibrillary tangles due to misfolded hyperphosphorylated tau in those neurons that innervate the PFC but not the hippocampus and that tau toxicity negatively impacted cognitive functions [105,106]. Other studies showed that depletion of noradrenergic LC neurons projecting into the hippocampus and the neocortex in a rodent AD model depleted hippocampal NE and impacted LTP and synaptic plasticity in this brain region [107]. Collectively, these studies indicate that NE that is released into the hippocampus from LC-producing neurons plays a role in hippocampal neurogenesis and in memory [108]. These findings led to the possibility of adopting a monoamine-based approach in the treatment of AD or in mitigating AD effects with aging. Table 2 summarizes the findings of select studies that demonstrated the role of adrenergic receptors in the brain.

## 4. Therapeutic Strategies for Alzheimer’s Disease and the Role of Epigenetics

Even though AD is one of the most prevalent neurodegenerative diseases worldwide, drugs that are currently used for AD treatment have failed to show long-term efficacy. AD is not only a disease that is due to genetic factors but also due to environmental factors, and epigenetic factors. Several strategic approaches to AD treatment exist. These approaches are “mechanism-based approaches and non-mechanism-based approaches” [109]. Accordingly, some drugs that manage the symptoms of AD have been approved by the FDA. It is obvious that we have a long way to go before curing this disease which suggests that other personalized treatments are needed at several stages of AD. These treatments should include not only drugs but also changes in lifestyle and diet in early life for better future outcomes.

Few studies have shown a correlation between the adrenergic system and epigenetic changes in modulating neuronal signaling. For example, an in vitro study showed that mouse hippocampal slices treated with NE led to activation of β-adrenergic receptors in CA1 neurons and triggered changes in DNA methylation and changes in histone marks (H3K14 acetylation and H3S10 phosphorylation) that resulted in LTP. The application of DNA inhibitor such as Azacytidine (AZA) to hippocampal slices resulted in a decrease in LTP. The application of an inhibitor of Aurora Kinase B that phosphorylates H3S10 resulted in a decrease in H3S10 phosphorylation. The application of an inhibitor of CBP/300 (a known histone acetyltransferase that causes histone acetylation) also reduced LTP [110]. Note that c-AMP response element binding protein (CREB), a known transcription factor that plays a role in long-term memory and synaptic plasticity, recruit CBP/300 [111]. In conclusion, this study showed that NE-induced LTP in mouse hippocampal slices required increased methylation and increased histone acetylation and phosphorylation [110]. Additional studies are needed to confirm the correlation between dysregulation of the adrenergic system (LC-NE) in AD to epigenetic dysregulation and whether this correlation is brain region-specific. Although many studies have demonstrated changes in several AD-related genes by epigenetic mechanisms [112], future studies should investigate if changes in cholinergic receptors expression in AD are mediated by epigenetic mechanisms.

The identification of biomarkers for AD diagnosis has been used as a standard in the clinical field. These biomarkers most often include key proteins that are malfunctioning, or their expression is dysregulated in neurons [113,114,115,116]. These key proteins could be Aβ and tau as well as other proteins that are markers for axonal degeneration, markers for synaptic degeneration, markers for neuronal injuries, markers for glial activation, markers for inflammation or markers for α-synuclein inclusions [117]. Most of the drugs used for AD treatment act as agonists or antagonists for specific types of cholinergic receptors or adrenergic receptors [55,118,119,120,121]. However, most of these drugs for AD did not show long-term efficacy [122]. Recent studies have shown that AD is caused by epigenetic mechanisms and suggested the use of the epigenetic machinery as a bridge that explains the interaction between genes and environmental factors in the etiology of AD [123,124]. Although AD has a genetic basis, recent studies indicate that environmental factors such as diet, lifestyle and social factors are linked to the etiology of AD [125,126]. More research should be done to better understand the role of gene–environment interaction in the etiology of neurodegenerative diseases. We will provide a brief overview on recent findings that linked epigenetic changes to AD pathology.

Epigenetic mechanisms play a role in gene expression regulation and have recently emerged as plausible mechanisms for understanding neurodegeneration [127]. The most widely studied epigenetic mechanisms are DNA methylation, histone modifications, chromatin-remodeling and the role of non-coding RNAs such as microRNAs. These mechanisms are mediated via the activity of enzymes such as DNA methyltransferases (DNMTs) or histone methyltransferases (HMTs). Figure 3 illustrates the role of epigenetic mechanisms in modulating the expression of many neuronal genes such as memory-related genes.

How are these epigenetic mechanisms linked to the pathology of AD and how can we use that knowledge in AD-related research and apply it clinically? AD neuropathology shows complex genetic and epigenetic components and complex interactions between the two factors suggesting that this neurodegenerative disease is impacted by environmental factors and shows global or gene-specific epigenetic alterations in neurons that impact their signaling [128]. Changes in DNA methylation of key neuronal genes related to aging and to the etiology of several neurodegenerative diseases have been reported suggesting the role of methylation in normal brain functioning and development throughout the lifespan [129]. Although most studies focused on the role of β-amyloid plaque deposition in AD pathogenesis, this by itself is not enough to understand the etiology as there are changes in methylation in AD that affect the expression of many Alzheimer’s-related genes. For example, methylome studies in AD brain demonstrated differentially methylated regions in several genes that are linked to β-amyloid plaque such as ANK1, RHBDF2, Bin1 and ABCA7 [130]. Hypomethylation of the APP gene promoter is an original finding that demonstrated the epigenetic basis of AD pathology and the contribution of DNA methylation in dysregulation of the APP gene expression in this disease [131]. Another study demonstrated the methylation of more than 27,578 CpG sites of several genes where TMEM59 gene plays an important role in amyloid-β-protein precursor post-translational processing, was found to be hypomethylated in the PFC of late-onset AD patients compared to normal controls [132]. Other studies showed changes in specific histone marks such as a decrease in the acetylation of H3K18/K23 in the temporal lobes of AD patients [133] and aberrant localization of the activation mark histone H3 trimethylation on lysine 4 (H3K4me3) in the early stages of the AD. This suggests a role of changes in histone marks in changes in the expression of synaptic genes in early AD [134].

The identification of tissue-specific epigenetic changes in the brain indicates the role of epigenetic mechanisms in the pathophysiology of AD. Other studies focused on understanding epigenetic changes in the blood of AD patients that act as biomarkers for early diagnosis of AD or for identification of those who are at risk of developing AD. For example, a global DNA hypermethylation was detected in peripheral blood mononuclear cells of late-onset AD patients compared to controls with a significant increase in the gene expression and protein levels of the maintenance DNA methyltransferase (DNMT1), an enzyme that causes and maintains the methylation of cytosine [135]. Interestingly, one study implicated the changes in the oxytocin gene methylation in the hypothalamus to AD etiology. The neuropeptide oxytocin is synthesized by hypothalamic neurons and has been shown to play a role in social cognition, behavior, and learning. They found differentially methylated region DMR in the proximity of the oxytocin promoter gene in the brain and the blood of AD patients suggesting that oxytocin gene and changes in its methylation maybe used as a biomarker in the early diagnosis of AD [136]. This study suggests that there is a state of hypermethylation of the oxytocin gene at the early stages of AD and a state of hypomethylation at late stages of AD. An elevation of oxytocin levels has been demonstrated in the hippocampus and the temporal cortex of postmortem AD brains and this elevation may be linked to memory impairment in AD [137]. This is quite important as AD patients have shown loss of paraventricular and supraoptic neurons of the hypothalamus that produce oxytocin [138].

Collectively, these studies clearly demonstrate that the etiology of AD causes modulation in gene expression by epigenetic mechanisms such as methylation. Targeting the epigenetic machinery responsible for DNA methylation via dietary supplementation of methyl-donors, which play a role in methylation, could have promising results [139,140,141,142]. Several reviews described the neuroprotective role of methyl-donors and DNA methylation status in neurodegenerative diseases [143,144,145].

## 5. Conclusions

The cholinergic system and the LC-NE system play important roles in cognitive functions such as attention, thinking abilities, learning and memory throughout the lifespan. The degeneration of cholinergic neurons and LC-NE neurons were reported in AD brains. This degeneration was associated with alteration in the levels of cholinergic or adrenergic markers that are essential in neuronal signaling. The modulation in the levels of Ach, NE, activity of cholinergic receptors and adrenergic receptors is also an important finding in AD research. These two neurotransmitters, Ach and NE, have been shown to have neuroprotective effects against neurodegeneration and their levels change at different stages of AD. Although most drug treatments for AD aimed at targeting specific molecular biomarkers implicated in this disease or aimed at targeting specific receptors such as AchRs and ARs, these drugs have failed to show long-term efficacy. Recently, there is a shift toward implementing more effective strategies and approaches for individualized and personalized treatments. These treatments do not only include drugs but also change in diet to affect the availability of methyl-donors. These later have been shown to alter DNA methylation and have shown promising effects in improving memory-related tasks in animal models of AD and in select AD patients. More studies are still needed to confirm the long-term beneficial effects of diet and lifestyle during early life in improving cognitive functions across the lifespan.

## Figures and Tables

**Figure 1 ijms-22-01273-f001:**
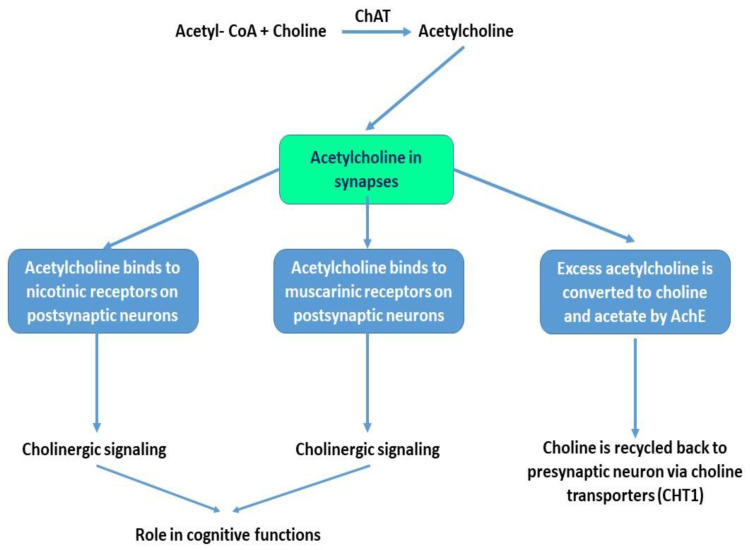
The role of acetylcholine at synapses. Choline is the main precursor for the formation of acetylcholine (Ach). Via the activity of choline acetyltransferase (ChAT), Ach is formed from choline and Acetyl-CoA. Once released at synapses, Ach binds to nicotinic or muscarinic receptors on postsynaptic neurons to regulate cholinergic signaling in different brain regions. Excess Ach at synapses is converted to choline and acetate via the activity of acetylcholine esterase (AchE). Choline is then recycled back to the presynaptic neuron via the presence of specific choline transporters such as CHT1.

**Figure 2 ijms-22-01273-f002:**
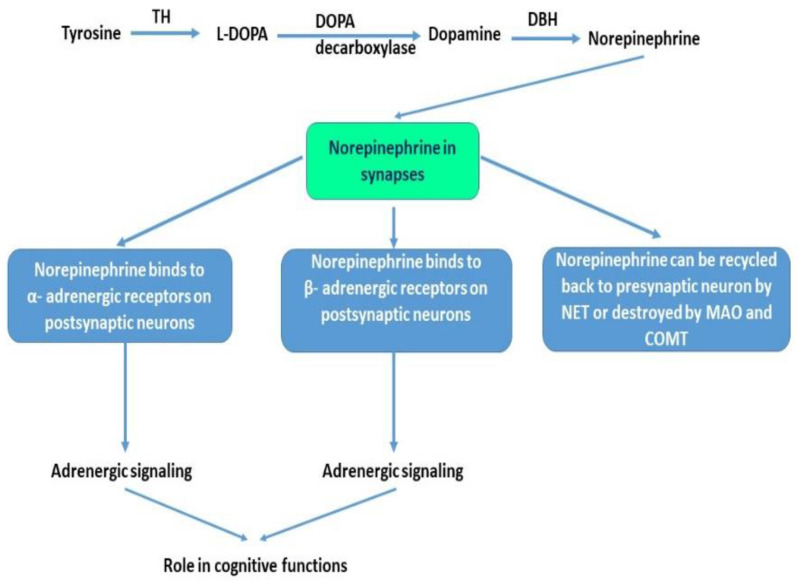
The role of norepinephrine at synapses. Norepinephrine is synthesized in a multistep process. Tyrosine is converted to L-DOPA via the activity of tyrosine hydroxylase (TH). L-DOPA is then decarboxylated via DOPA decarboxylase into dopamine. Dopamine via dopamine beta-hydroxylase (DBH) is then converted to norepinephrine (NE). NE is released at synapses where it can bind to α- adrenergic receptors or β- adrenergic receptors on postsynaptic neurons to modulate neuronal firing or modulate adrenergic neurons signaling. NE excess can be eliminated via the activity of monoamine oxidase (MAO) or catechol-O-methyltransferase (COMT). NE can also be recycled back to presynaptic neurons via the NE transporters (NET).

**Figure 3 ijms-22-01273-f003:**
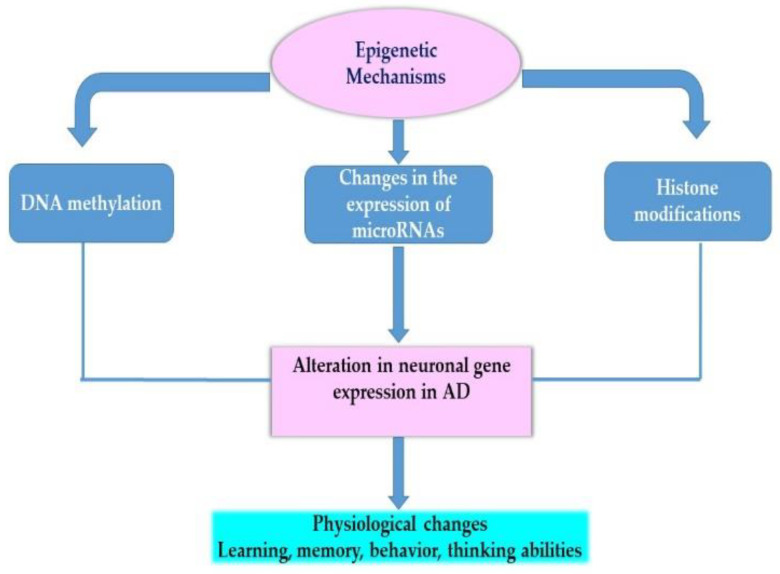
The role of epigenetic mechanisms in modulation of gene expression. This figure provides an overview of the role of epigenetic mechanisms in modulation of the genome and the consequences of such modulation on phenotype. These mechanisms include DNA methylation of genes, changes in specific histone marks along genes or changes in the expression of genes by specific microRNAs.

**Table 1 ijms-22-01273-t001:** Summary of the role of cholinergic receptors in the brain. The role of the cholinergic receptors such as nicotinic (nAchRs) or muscarinic (mAchRs) receptors in different brain regions of animal models is summarized.

Description	Outcomes	References
α7 nAchRs	Enhancement of LTP in the hippocampus and in the amygdala	[46,47]
3xTg-AD mouse model	Reduction in α7 nAchRs levels in an age-dependent manner	[49]
Blockade of mAchR signaling by an antagonist in rats	Impairment in the acquisition and consolidation of contextual fear conditioning	[51]
Muscarinic agonist against M1 mAchRs in a 3xTg-AD mouse model	Decrease in β-amyloid plaques accumulation, decrease in tau hyperphosphorylation, improvement in cholinergic activity in the cortex and the hippocampus and improved some cognitive functions	[54]
Knockout of M1 and M4 mAchRs in mice	Impacted hippocampal circuitry and hippocampal release of Ach	[57,58]
Lack of M1 AchR in mouse PFC	Impairment of cholinergic signaling in pyramidal neurons and cue detection deficits	[59]
M2 AchR knockout mice	Deficit in specific learning and memory tests	[62]

**Table 2 ijms-22-01273-t002:** Role of adrenergic receptors in the brain. The role of adrenergic receptors such as α or β adrenergic receptors (ARs) in different brain regions of Alzheimer’s disease (AD) mice models is summarized.

Description	Outcomes or Role	References
Blockade of α1 ARs in APP23 transgenic mice model of AD	Impairment in cognitive functions	[94]
β2 AR in PFC of APP/PS1 transgenic mouse model	Alteration in Amyloid Precursor Protein (APP) processing and mediation of Aβ-induced tau pathology	[97]
α2A AR in cerebral cortex of AD transgenic mice model	Regulation of APP proteolytic processing and Aβ production and secretion	[98]
Blockade of β ARs in APP mouse model	Impairment in cognitive functions, and behavior. Inducing inflammation with chronic blockade of these receptors	[103]
β1 AR activation in hippocampal pyramidal neurons in mice	Role in contextual and spatial memory consolidation and retrieval	[100,101,102]

## Data Availability

Not applicable.

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
