# Peer review of "The Cholinergic System, the Adrenergic System and the Neuropathology of Alzheimer’s Disease"

_ijms, 2021, doi:10.3390/ijms22031273_

Round 1

Reviewer 1 Report

Reviewer’s comments and suggestions

The goal of this review was to discuss the latest literature on Alzheimer's disease pathology and possible therapeutic effects of the cholinergic system and the adrenergic system (LC-NE) system. The author, however, must write about the design of writing this review, I mean he should discuss a single line in the abstract's material and process section rather than skipping an important section. It should be specific n terms of selection as well. People know that these systems are involved in pathology but the author should have to provide novelty in the paper and reported some interesting finding in the paper. The author should provide 2-3 figures and discuss them well. The sections were good but need to thorough revise your paper based on my comments.

Decision: Major revision

  1. Please avoid using big sentences, line number 39-43
  2. Line 57 and 60, the sentences used we will was not appropriate to use
  3. The author should write the legend part in both tables
    4. Line number 97-98, need to be discuss
  4. Line 116-117 already discussed
  5. Line 149, please discuss the relevant studies
  6. Please check line 171-173, line 194, you need to provide details if it is good or bad
  7. Line 198, Nerve growth factor needs to be incorporated
  8. Please check the journal style of the IJMS in the manuscript text.
  9. Line 209, 293needs a reference for this line
  10. Line 217-220 The line seems to be done experimented by you, if not change the sentence into an appropriate one
  11. Please avoid this kind of sentences in review (line number 306-307,363, 437-438)
  12. Line 368, you can cite some relevant references such as https://jamanetwork.com/journals/jamaneurology/fullarticle/2758654

https://www.sciencedirect.com/science/article/pii/S266614461930005X

https://www.nature.com/articles/s41380-020-0721-9

https://pubmed.ncbi.nlm.nih.gov/32392110/

Author Response

Reviewer 1:

Thank you Reviewer 1 for the reviews of my paper (#ijms-1070151).  The reviews were very useful to me in making revisions to the manuscript, and I hope that it will now be suitable for publication.  Please find below my reply to the reviewers in bold.

1. Please avoid using big sentences, line number 39-43
The sentence is now revised and split into 2 sentences (Lines 39-41 and lines 41-44).

2. Line 57 and 60, the sentences used we will was not appropriate to use
The sentence is now revised (line 57).

3. The author should write the legend part in both tables
Legend is now added to tables 1 & 2 (Lines 248-250 and 383-384).

4. Line number 97-98, need to be discuss

I revised this information in lines 104-109 and added references 13-16).

5. Line 116-117 already discussed --- This sentence is now deleted (Lines 125-126).

6. Line 149, please discuss the relevant studies
The information about α4β2 nAchRs is explained in lines 160-164 and cited.

7. Please check line 171-173,

This sentence is now revised (Lines 183-185).

line 194, you need to provide details if it is good or bad
An explanation is now added (Lines 208-209).

8. Line 198, Nerve growth factor needs to be incorporated

Nerve Growth Factor is now added (Line 212).

9. Please check the journal style of the IJMS in the manuscript text.

I did follow the journal requirements in writing this manuscript and the style was adjusted by the journal editorial office.

10. Line 209, 293 needs a reference for this line
Reference 53 is now added (Line 225)
Reference 88 is now added (Line 321)

11. Line 217-220 The line seems to be done experimented by you, if not change the sentence into an appropriate one

Reference 56 is now added (Line 236).

Please avoid this kind of sentences in review (line number 306-307,363, 437-438)

306-307: The sentence is revised (Line 333- 334).
363:  This sentence is deleted.  I cited Reference 111 (Line 390/393).
437-438:  The sentence is revised (Line 496).

12. Line 368, you can cite some relevant references such as https://jamanetwork.com/journals/jamaneurology/fullarticle/2758654

https://www.sciencedirect.com/science/article/pii/S266614461930005X

https://www.nature.com/articles/s41380-020-0721-9

https://pubmed.ncbi.nlm.nih.gov/32392110/

Thank you for suggesting these references.  References 115-118 are now added (Line 419).

13. Per Reviewer 1 request, I added 3 figures with explanation to the manuscript. Please refer to lines 97-102 on page 3, 287-295 on page 7, and 441-446 on page 11.

Reviewer 2 Report

Given that there is a plethora of papers on Alzheimer’s disease (AD), it is a monumental task for students, researchers and scientists to examine in detail every single new paper relevant to their work and research interest. A well written scientific review that summarize the research work and discusses it critically, identifies methodological problems, provides alternate outcomes, points out pitfalls and research gaps is not only informative but also provides the reader with new and meaningful ideas and a path to follow.

This review article “The cholinergic system, the adrenergic system and the neuropathology of Alzheimer’s disease” is well written and provides good insight and elaborate information on one aspect of AD: the cholinergic and the adrenergic system

I have few queries which if addressed will improve the manuscript.

  1. On page 3, Lines 142-144, the author states that a reduction in the density of α7nAchRs may play a significant role in AD pathogenesis. However, on page 4, lines 171-173, it is pointed out that elevated levels of α7nAchR negatively impacts cholinergic signaling and cognitive functions. Both observations are contradictory.

  1. Throughout the review the author discusses the reduction in the expression or activity of AchE or NE receptors that may have a role in AD pathogenesis. Can the author state in few lines what receptor reduction means? Is it actual decrease in the receptor number, or is it receptor desensitization or downregulation of receptor activity? What is the cause for this phenomenon?

  1. Could the reduction in Ach and NE activity or a reduction in receptor reduction be a post-mortem effect?

  1. The author needs to provide some information on the cholinergic AD drug that was approved but which has limited efficacy and use.

  1. Are the cholinergic and adrenergic systems subject to epigenetic changes and/or methylation changes? How does that modification affect downstream signaling in relation to AD?

Author Response

Reviewer 2:

Given that there is a plethora of papers on Alzheimer’s disease (AD), it is a monumental task for students, researchers and scientists to examine in detail every single new paper relevant to their work and research interest. A well written scientific review that summarize the research work and discusses it critically, identifies methodological problems, provides alternate outcomes, points out pitfalls and research gaps is not only informative but also provides the reader with new and meaningful ideas and a path to follow.

This review article “The cholinergic system, the adrenergic system and the neuropathology of Alzheimer’s disease” is well written and provides good insight and elaborate information on one aspect of AD: the cholinergic and the adrenergic system

I have few queries which if addressed will improve the manuscript.

Thank you Reviewer 2 for the reviews of my paper (#ijms-1070151).  The reviews were very useful to me in making revisions to the manuscript, and I hope that it will now be suitable for publication.  Please find below my reply to the reviewers in bold.

1. On page 3, Lines 142-144, the author states that a reduction in the density of α7nAchRs may play a significant role in AD pathogenesis. However, on page 4, lines 171-173, it is pointed out that elevated levels of α7nAchR negatively impacts cholinergic signaling and cognitive functions. Both observations are contradictory.

I revised the sentence to indicate that a decrease in α7 nAchRs - Aβ complex formation have been shown to improve learning and memory (lines 183-186).

2. Throughout the review the author discusses the reduction in the expression or activity of AchE or NE receptors that may have a role in AD pathogenesis. Can the author state in few lines what receptor reduction means? Is it actual decrease in the receptor number, or is it receptor desensitization or downregulation of receptor activity? What is the cause for this phenomenon?

Some studies showed a reduction in the number of receptors (protein levels) which leads to a reduction in activity.  Other studies documented a change in gene expression which does not necessarily means a reduction in activity.  A reduction in activity could be due to desensitization of receptors or decrease in the number of the receptors (protein levels).  The desensitization is due to persistent elevated levels of Ach or NE.   I revised the information in lines 144 and 163.

  1. Could the reduction in Ach and NE activity or a reduction in receptor reduction be a post-mortem effect

I agree with Reviewer 2 that a reduction in neurotransmitter function or a reduction in the number of receptors could be a postmortem event.  Several studies have shown that such reduction is not the same in patients at different stages of AD.  So, it is pertinent in future studies to compare changes in the activity/number of receptors between AD brain and AD postmortem brain to come up with better conclusion. 

4. The author needs to provide some information on the cholinergic AD drug that was approved but which has limited efficacy and use.

Reference 124 is now added (Line 425).

5. Are the cholinergic and adrenergic systems subject to epigenetic changes and/or methylation changes? How does that modification affect downstream signaling in relation to AD?

I have summarized a study that showed a correlation between the adrenergic system and induction of epigenetic changes (Lines 398-415).

Reviewer 3 Report

The review is well written and presents pertinent information about the cholinergic the adrenergic systems and their role in pathophysiology and treatment of AD. The review is appropriately structured and adequately covers the scope of the problem outlined in the introduction. I have a few minor suggestions/edits, outlined below, and once they are addressed the review can be accepted for publication.   

Few edits - 

Lines 97-98 : Rephrase as 'This hypothesis is still largely accepted by the scientific community, however, it has been challenged by some
and several other competing hypotheses have emerged' and provide reference(s).

Lines 208-210 - provide reference(s)

Line 362 - rephrase as '..some drugs that manage the symptoms
of AD have been approved by FDA.'

Author Response

Reviewer 3:

The review is well written and presents pertinent information about the cholinergic the adrenergic systems and their role in pathophysiology and treatment of AD. The review is appropriately structured and adequately covers the scope of the problem outlined in the introduction. I have a few minor suggestions/edits, outlined below, and once they are addressed the review can be accepted for publication.   

Thank you Reviewer 3 for the reviews of my paper (#ijms-1070151).  The reviews were very useful to me in making revisions to the manuscript, and I hope that it will now be suitable for publication.  Please find below my reply to the reviewers in bold.

  1. Lines 97-98 : Rephrase as 'This hypothesis is still largely accepted by the scientific community, however, it has been challenged by some and several other competing hypotheses have emerged' and provide reference(s).

This sentence is now revised (Lines 104-109) and references 13-16 are now added.

  1. Lines 208-210 - provide reference(s)

Reference 53 is now added (Line 225).

  1. Line 362 - rephrase as '..some drugs that manage the symptoms
    of AD have been approved by FDA.'

This sentence is now rephrased as suggested by Reviewer 3 (Lines 393-394).

Round 2

Reviewer 1 Report

No more comments.